# Heterogenization of Heteropolyacid with Metal-Based Alumina Supports for the Guaiacol Gas-Phase Hydrodeoxygenation

**DOI:** 10.3390/molecules28052245

**Published:** 2023-02-28

**Authors:** Rita F. Nunes, Daniel Costa, Ana M. Ferraria, Ana M. Botelho do Rego, Filipa Ribeiro, Ângela Martins, Auguste Fernandes

**Affiliations:** 1Centro de Química Estrutural, Institute of Molecular Sciences, Universidade de Lisboa, Av. Rovisco Pais, 1049-001 Lisboa, Portugal; 2Centro de Química Estrutural, Instituto Superior Técnico, Universidade de Lisboa, Av. Rovisco Pais, 1049-001 Lisboa, Portugal; 3BSIRG-iBB-Institute for Bioengineering and Biosciences, Chemical Engineering Department, Universidade de Lisboa, 1049-001 Lisbon, Portugal; 4Associate Laboratory i4HB—Institute for Health and Bioeconomy, Instituto Superior Técnico, Universidade de Lisboa, 1049-001 Lisbon, Portugal; 5DEQ, Instituto Superior de Engenharia de Lisboa, Instituto Politécnico de Lisboa, Rua Conselheiro Emídio Navarro, 1959-007 Lisboa, Portugal; 6Centro de Química Estrutural, Institute of Molecular Sciences, Faculdade de Ciências, Universidade de Lisboa, Ed. C8, Campo Grande, 1749-016 Lisboa, Portugal

**Keywords:** biomass valorization, hydrodeoxygenation, heteropolyacid, heteropolyacid salt, bifunctional catalysts, guaiacol

## Abstract

Because of the global necessity to decrease CO_2_ emissions, biomass-based fuels have become an interesting option to explore; although, bio-oils need to be upgraded, for example, by catalytic hydrodeoxygenation (HDO), to reduce oxygen content. This reaction generally requires bifunctional catalysts with both metal and acid sites. For that purpose, Pt-Al_2_O_3_ and Ni-Al_2_O_3_ catalysts containing heteropolyacids (HPA) were prepared. HPAs were added by two different methods: the impregnation of a H_3_PW_12_O_40_ solution onto the support and a physical mixture of the support with Cs_2.5_H_0.5_PW_12_O_40_. The catalysts were characterized by powder X-ray diffraction, Infrared, UV-Vis, Raman, X-ray photoelectron spectroscopy and NH_3_-TPD experiments. The presence of H_3_PW_12_O_40_ was confirmed by Raman, UV-Vis and X-ray photoelectron spectroscopy, while the presence of Cs_2.5_H_0.5_PW_12_O_40_ was confirmed by all of the techniques. However, HPW was shown to strongly interact with the supports, especially in the case of Pt-Al_2_O_3_. These catalysts were tested in the HDO of guaiacol, at 300 °C, under H_2_ and at atmospheric pressure. Ni-based catalysts led to higher conversion and selectivity to deoxygenated compound values, such as benzene. This is attributed to both a higher metal and acidic contents of these catalysts. Among all tested catalysts, HPW/Ni-Al_2_O_3_ was shown to be the most promising, although it suffered a more severe deactivation with time-on-stream.

## 1. Introduction

Many environmental issues have been raised over the years because of the intensive use of fossil fuels, which accounts for around 80% of the primary energy use [1]. Moreover, around 90% of CO_2_ emissions are attributed to the burning of fossil fuels [2]. Although carbon emissions must be mitigated, it is unlikely that it will be done through a reduction in energy production, mainly because the energy demand is growing [2]. This means that energy production must be approached in a more effective way. To address this issue, carbon-neutral energy sources are becoming increasingly popular, with biomass being one example as it is carbon neutral, renewable and available [3]. To achieve a workable starting material, biomass must go through pyrolysis to produce bio-oil [4]. However, this bio-oil composition is far from the desired for fuel, mainly due to the presence of oxygenated molecules, amongst other problems [5,6]. Thus, an upgrade is needed and is usually done by catalytic hydrodeoxygenation (HDO). This catalysis is usually heterogeneous due to, amongst other reasons, easy separation from the reaction medium and more stability at higher pressures and temperatures [7]. Conventionally, HDO accompanies the hydrodesulfurization (HDS) process, whose operating conditions are very harsh. Catalysts are only active in their sulfide form, which is a problem for the HDO process since sulfur from the catalysts is a source of contamination for the bio-oil stream [1]. Thus, new heterogenous catalysts, operating at lower pressures and without the presence of sulfur, are currently the focus of many HDO research studies. HDO catalysts must be carefully designed, attending to the needs of this reaction. A metal function is needed for hydrogenation purposes as well as the C-C bond cleavage. In addition, an acidic function complements the first one once it is where the C-O activation occurs. The acidity function is usually provided by the support, where the metal is then added. Examples of catalysts are metals with hydrogenation capacity, such as Pt or Ni added to acidic supports, such as γ-Al_2_O_3_ (which is amphoteric by nature [8]) or HZSM-5 [9,10]. Carbon-supported catalysts are also an option, due to their high surface area. As well as aluminas [8], carbon supports can have a basic or acidic nature [11]. Thus, the acidic function can be provided by heteropolyacids (HPA) and their salts, whether as a support or impregnated onto other supports. The HDO reaction, comprising HPA, has been studied for the removal of oxygen from molecules such as transanethole with Ni and HPW supported on SBA-15 [12], ketones with Pt and Cs_2.5_ salt [13] and anisole with phosphomolybdic acid on TiO_2_ [14].

HPA heterogenization has been the subject of many studies [3,15,16]. For example, HPA dispersion onto an inorganic support with a high specific surface area can bring interesting features to the final heterogenous catalysts. One example is supporting HPA on SiO_2_ for the esterification of acetic acid, in the presence of 1-heptanol for heptyl acetate production [7]. However, although transition-phase aluminas can present excellent features (large surface area, thermal stability for instance) and are often used as supports for many catalytic systems [17], in the case of HPA immobilization, some decomposition, or at least a strong interaction between acidic HPA and the basic sites of the support might be expected in this particular case [15,16]. 

In this work, different heteropolyacid-based heterogeneous catalysts were prepared by using alumina support. In order to study the interaction between the HPA and the Al_2_O_3_ support, two strategies were considered for the preparation of the solid catalysts: (a) impregnation of H_3_PW_12_O_40_ (HPW) onto an Al_2_O_3_ matrix previously modified with Ni and (b) mechanical mixture of Al_2_O_3_ support with HPW previously exchanged with Cs in order to obtain an insoluble Cs_2.5_H_0.5_PW_12_O_40_ salt. All of the solid catalysts obtained were tested in guaiacol HDO (at 300 °C under atmospheric pressure) and the two different preparation strategies were compared. 

## 2. Results

Hereafter, the results concern the physicochemical characterizations of the different HPW or Cs_2.5_ salt-modified catalysts and their comparison with raw supports, namely Pt-Al_2_O_3_ (1 wt.% Pt) and Ni-Al_2_O_3_ (28 wt.% Ni). Figure 1 shows an insight into the different catalysts prepared. For information, textural parameters of the raw supports are: (a) Pt-Al_2_O_3_ V_p_ = 0.45 cm^3^.g^−1^, S_BET_ = 155 m^2^.g^−1^; (b) Ni-Al_2_O_3_ V_p_ = 0.49 cm^3^.g^−1^, S_BET_ = 215 m^2^.g^−1^. H_2_/O_2_ titration experiment performed onto raw Pt-Al_2_O_3_ support gave a metal dispersion of 35%, confirming a highly dispersed metallic Pt phase. On the other side, H_2_-TPR experiments performed on the Ni-Al_2_O_3_ phase revealed Ni species with different reducibility: on one hand, NiO species were easily reducible (peak maximum at 200 °C) and, on the other hand, NiO species interacting strongly with the alumina support, with a reduction peak at about 450 °C. Nevertheless, as catalysts were all pretreated at about 300 °C under H_2_ atmosphere (reaction conditions) before catalytic evaluation, one can assume a Pt phase is almost metallic in nature for Pt-Al_2_O_3_ support-based catalysts and, on the other hand, oxidized Ni species together with metallic Ni ones for Ni-Al_2_O_3_ catalysts during catalytic evaluation. 

### 2.1. Catalyst Characterization

#### 2.1.1. Structural Properties

The Powder X-ray diffractograms (PXRD) for all the catalysts in the study can be seen in Figure 1. For each series, all the catalysts are compared with the respective support, Pt-Al_2_O_3_ and Ni-Al_2_O_3_, and also the Cs_2.5_ salt compound. The PXRD of the HPW material is shown in Appendix A. 

Regarding the Pt-Al_2_O_3_-based catalysts, in Figure 1A, one can see that raw Pt-Al_2_O_3_ support is presenting an XRD pattern corresponding fairly to γ alumina (COD number 1101168), with no peaks from the platinum phase. This is seemingly because the amount of the latter is very low (1 wt.%) and the metal is well dispersed onto the support, confirming the dispersion of the 35% obtained from the H_2_/O_2_ titration experiment. 

On the other hand, raw-Ni-Al_2_O_3_ support presents a similar PXRD pattern as the Pt-Al_2_O_3_ one. Additionally, no peaks from the reduced or oxidized Ni phases seem to be present, thus showing that the Ni species are well dispersed onto the alumina support. In both cases, the diffraction peaks of both supports are low in intensity and also very broad, probably because alumina crystallites are very small in size.

Concerning the impregnated heteropolyacid/alumina support samples (HPW/Pt-Al_2_O_3_ and HPW/Ni-Al_2_O_3_), in both cases, XRD patterns resemble the support ones, although with peaks of lower intensity. Indeed, no peaks of bulk HPW heteropolyacid can be observed. This is probably because a good HPW dispersion has been achieved during sample preparation. Other authors also reported the absence of peaks coming from bulk HPA, when supporting heteropolyacid onto oxide support. For example, Caliman et al. [18] only observed diffraction peaks from hydrated HPW when the amount of the heteropolyacid was 80 wt.% or higher. On the other hand, Liu et al. [19] could not identify by XRD peaks from phosphotungstic acid on their final samples, assuming that this acid was well dispersed onto the Ni-Al_2_O_3_ support, probably as an amorphous phase. Forster et al. [20] reported the same observation when using either γ- or θ-alumina support. 

On the other hand, for the physically mixed catalysts (Pt-Al_2_O_3_/Cs_2.5_ and Ni-Al_2_O_3_/Cs_2.5_), XRD patterns for both catalysts look very similar. In this case, experimental patterns can be seen as the sum of individual contributions of the support and Cs_2.5_ salt, although the peaks of the supports are very low in intensity. Here, the diffraction pattern of Cs_2.5_ salt might prevail over the support one, for some reasons: Cs_2.5_ salt is more crystalline than the support, difference in absorptivity, etc. However, together with the intense peaks from the Cs_2.5_ salt phase, one can also observe the main peaks of both Ni and Pt-Al_2_O_3_ supports at about 37 and 45° (2 theta) (see Figure 1).

#### 2.1.2. Spectroscopic Properties

IR spectra of the different samples are presented in Figure 2. Concerning both the support bands, Pt- and Ni-Al_2_O_3_, at 1360 (Al-OH stretching), 1060 (Al-OH bending), 810 and 740 cm^−1^ (AlO_6_ and AlO_4_ stretching, respectively), characteristic of γ alumina support, can be observed [21]. Interestingly, in the case of HPW-impregnated samples, IR spectra are very similar to pure supports ones, meaning that no traces of the Keggin anion can be detected. Caliman and coworkers [18] also verified, by Infrared, that characteristic bands from HPW were only observed for heteropolyacid content of about 40 wt.%. which, according to the authors, corresponds to the formation of a monolayer at the surface of the alumina support. The authors also concluded that below that content, the present HPW species might be well dispersed or even decomposed because of the strong interaction with the support. On the other hand, for physically mixed samples, the bands from Cs_2.5_ salt are logically visible, at about 1080, 986, 887 and 808 cm^−1^ and correspond to pure Cs_2.5_ salt compound (as observed in Figure 2), which is also very similar to HPW compound [22] (see spectrum of HPW compound in Appendix A). 

Raman spectroscopy was carried out for all the samples, as well as for starting HPW and Cs_2.5_ salt compounds. The results are depicted in Figure 3. For Pt-and Ni-Al_2_O_3_ samples, no bands from the alumina support are visible but only some fluorescence from the support itself. On the other hand, for all the supported HPW and Cs_2.5_ salt samples, a band at ca. 1000 cm^−1^, characteristic of the Keggin anion, can be observed [20]. Other bands at about 500, 200 and 100 are also visible for samples Pt-Al_2_O_3_/Cs_2.5_, Ni-Al_2_O_3_/Cs_2.5_ and HPW/Ni-Al_2_O_3_, meaning that both HPW and Cs_2.5_ salt are present on the alumina support, although well dispersed because bands are relatively broad. On the opposite, the Raman spectrum of HPW/Pt-Al_2_O_3_ sample presents a very poorly resolved peak. Here, it seems that HPW interacts more strongly with Pt-Al_2_O_3_ support, resulting in a weaker Keggin anion signal. The difference in the behavior between HPW impregnated in Pt or Ni support might be due to the metal content. Once Ni-Al_2_O_3_ catalysts have a much higher metal content than Pt ones, this metal may act as a passivating agent, preventing a stronger interaction between HPW and alumina surface. 

UV-Vis DRS spectra of the different catalysts are presented in Figure 4, together with HPW and Cs_2.5_ salt. On the left side, one can see the spectra of Pt-Al_2_O_3_-based catalysts (Figure 4A). The support itself does not show any characteristic band, but only a very broad massif at 200–400 nm (maximum at 220 nm), typical of LMCT (ligand-to-metal charge transfer) bands in metal oxides. The spectra of HPW/Pt-Al_2_O_3_ and Pt-Al_2_O_3_/Cs_2.5_ samples are, on the other hand, dominated by an intense band at 259 and 264 nm, respectively. This band corresponds to the respective heteropolyacid (HPW and Cs_2.5_ salt), although with a little red shift of the wavelength (maximum is at 255 nm for both HPW and Cs_2.5_ salt materials). Interestingly, the shoulder clearly observable at about 305 nm for the raw HPW sample (Figure 4C) does not appear in the spectrum of HPW/Pt-Al_2_O_3_ sample. The same trend was also observed by Sudhakar et al. [23]. A possible explanation could be the loss of long-range order of HPW structure, resulting in the disappearance of the respective shoulder. In Figure 4B (right side), spectra of the series Ni-Al_2_O_3_ are shown. Ni-Al_2_O_3_ support presents, together with the large LMCT band_,_ additional bands at 400, 560 and 620 nm, probably from the presence of NiO and Ni species at the surface of the alumina support. As a consequence, the highly absorbing sample presents a black color. HPW/Ni-Al_2_O_3_ sample presents the same bands as Ni-Al_2_O_3_ one. However, it also presents an additional absorption contribution below 250 nm (baseline is higher than that of the pure Ni-Al_2_O_3_ support), possibly coming from HPW species. Finally, Ni-Al_2_O_3_/Cs_2.5_ sample also presents both contributions, a large band from the support (more pronounced for Ni-Al_2_O_3/_Cs_2.5_ sample) and a maximum at 266 nm coming from the Cs_2.5_ salt component (see Figure 4D). 

#### 2.1.3. XPS Results

The following sections describes the XPS results obtained for all the six catalysts, Figure 5, Figure 6, Figure 7 and Figure 8 show the different XPS spectra obtained (W 4f, P 2p, Al 2s and 2p and finally Ni 2p regions). Appendix A gives the experimental atomic ratios obtained from XPS experiments and compared with the theoretical ones. Apart from hydrogen, all the elements of the heteropolyacid and salt were detected in all the samples. For the samples containing the Cs_2.5_ salt, the theoretical values are generally in agreement with experimental ones, which is not always the case for HPW samples. This is probably because the Cs_2.5_ salt samples were prepared by physical mixture whereas HPW ones were prepared by impregnation. In the first case, the Cs_2.5_ salt compound is present but not modified while in the second case, HPW interacts strongly with the alumina support. Therefore, some attenuation of the HPW signal may occur, which could alter slightly the experimental atomic ratios expected. 

In Figure 5, one can see the W 4f region for the different catalysts. HPW and Cs_2.5_ salt samples show the typical W 4f doublet with a spin-orbit separation of 2.1 eV and the main component, W 4f_7/2_, centered at 36.5 ± 0.2 eV and assigned to W(VI) in the heteropolyacid [24]. In the case of mechanical mixtures (Pt-Al_2_O_3_/Cs_2.5_ and Ni-Al_2_O_3_/Cs_2.5_), the same doublet also appears, but less intense, because of the dilution effect (20 wt.% Cs_2.5_ salt in both supports). On the other side, for HPW impregnated onto the two different supports, W 4f signal is much less intense than in the mechanical mixtures which shows that the HPA is buried in the pores of the support. To obtain a good fit of the overall signal, two doublet peaks are needed, with W 4f_7/2_ fitted peaks centered at 35.6 ± 0.3 eV and 36.5 ± 0.3 eV. Both doublets may be attributed to W(VI) [24], but probably originating from different chemical environments. These two observations allow one to conclude that, contrary to what happens for Cs_2.5_ salt mixtures, when HPW is introduced by impregnation, a strong interaction between HPW and the respective support occurs.

In fact, when looking at the spectra of the different catalysts in the P 2p region (Figure 6), the same conclusions can be also found. P 2p is also composed of a doublet peak with spin-orbit separation of 0.87 eV, being the main component, P 2p_3/2_, centered at 134.6 ± 0.1 eV, consistent with PO_4_^2−^ from the heteropolyacid. P 2p regions are strongly attenuated for the samples of HPW impregnated in Al_2_O_3_-based supports (both with Pt and Ni, spectra c and e) and are also broader, revealing once again, the strong interaction between HPW and the support. On the contrary, for Cs_2.5_-based samples (spectra d and f), P 2p region peaks are very similar to the respective P 2p from HPW and Cs_2.5_ salt compounds, showing that Cs_2.5_ salt structure is logically well preserved in the final physical mixture.

Figure 7 presents the spectra obtained for the different samples, in the Al 2s and Al 2p regions. For sake of simplicity, and because many XPS regions overlap (see Section 3.2 for detail), Al 2s region was used for the Pt-Al_2_O_3_-based samples, while Al 2p region was used for the Ni-Al_2_O_3_ ones. In the case of Pt-Al_2_O_3_ samples (spectra a, b and c), the Al 2s region (singlet) was fitted with just one peak centered at 120.0 ± 0.1 eV for the three samples, typical of Al^3+^ from Al_2_O_3_. In the case of the Pt-Al_2_O_3_/Cs_2.5_ sample, the signal is less intense because of the dilution effect. Concerning the Ni-Al_2_O_3_-based samples (spectra d, e and f), the doublet from Al 2p has a spin-orbit separation of 0.41 eV and the main component Al 2p_3/2_ is centered at 74.7 ± 0.1 eV, being assigned to Al_2_O_3_. The superpositions found in each region are identified. In Ni-Al_2_O_3_/Cs_2.5_, Ni 3p also exists (as attested by the Ni 2p region, which is clearly detected, as shown below); however, it has a very low intensity, and therefore, no attempts to fit the region nor to quantify it were performed. Here again, for the Ni-Al_2_O_3_/Cs_2.5_ sample (obtained from a mechanical mixture), the Al signal is less intense when compared with the two other samples (Ni-Al_2_O_3_/Cs_2.5_ and HPW/Ni-Al_2_O_3_). 

Finally, Figure 8 presents the XPS spectra for the Ni-Al_2_O_3_ catalysts series, showing the Ni 2p region. Ni 2p is also a doublet peak with an associated multiple structure, next to each photoelectron peak (at higher BE), arising from the presence of unpaired electrons in the valence band. All the catalysts present a similar spectrum that fits very well with the spectrum of the NiO(OH) species [25]. In the case of the Ni-Al_2_O_3_/Cs_2.5_ sample, the respective spectrum is somewhat less intense than for the two other samples, due to the matrix dilution effect. 

#### 2.1.4. Acidity Properties

Figure 9 shows the different NH_3_-TPD profiles obtained for each catalyst. All the materials present a profile with two main peaks, one at low-temperature LT (with a maximum at about 260 °C) and another at high-temperature HT (ca. 430–460 °C). These two peaks correspond to, respectively, weak and strong acid sites. The NH_3_-TPD profiles of the reference materials are presented in Appendix A for comparison. 

Table 1 summarizes the amount of NH_3_ desorbed for each catalyst and each peak. Concerning the raw supports, one can see that both present a large amount of weak acid sites (about 60–70%). However, Ni-Al_2_O_3_ support shows a larger amount of total acid sites when compared with Pt-Al_2_O_3_. This can be explained by the fact that: a) Ni-Al_2_O_3_ presents a much larger surface area when compared with Pt-Al_2_O_3_ one (215 against 155 m^2^.g^−1^); the presence of oxygenated Ni species might also account for additional Lewis acid sites. When comparing both series’, interesting findings can be seen. For example, Pt-Al_2_O_3_-based catalysts present a rather low total acidity (<285 μmol·g^−1^), whereas Ni-Al_2_O_3_-based catalysts present a higher total acidity (>340 μmol·g^−1^). While raw supports present essentially weak acid sites, HPW- and Cs_2.5_-based samples now present a relatively higher amount of strong acid sites. 

Yet, as NH_3_-TPD experiments do not allow us to differentiate between Brönsted and Lewis acid sites, it is difficult to say if our samples present both types of acidity. Nevertheless, an interesting study from Nowinska et al. might give some insights into the nature of the acid sites in our samples [26], as their HPA-alumina materials preparation is very similar to ours (impregnation method followed by calcination at about 400 °C). According to them, when heteropolyacids are impregnated onto oxide support with basic properties (such as Al_2_O_3_), they interact strongly with the latter and aluminum salts of the respective HPA can be formed. In that case, and when the amount of HPA is low enough to avoid a complete coverage of the support surface (that occurs at around 40 wt.% of HPW [18]), the heteropolyacid does not show Brönsted acidity any longer. Only an increase in HPA at the surface of the support could allow it to overcome the formation of such Al salts, giving rise to the appearance of Brönsted acid sites at the surface of the heterogenous catalysts. This appears when the HPA amount exceeds the formation of a complete monolayer. In our case, the amount of HPW impregnated is 20 wt.%, which is below the amount required to form a monolayer of Keggin ions at the surface of alumina support [18]. In that case, a large part of the HPA species might have been converted into those aluminum salt species, with no Brönsted acid sites, but instead Lewis acid sites. Indeed, it is well known that metal salts of HPA potentially show Lewis acidity, originating from the metal cations themselves, that behave as electron pair acceptors [16]. In summary, the presence of such aluminum salts of the respective HPW compound, might explain, on one hand, why our samples present a better acidity than the respective raw supports and, on the other hand, why HPW interacts so strongly with alumina support in a manner that HPW fingerprints seem to vanish, as evidenced by all the results we obtained (XPS, XRD, DRS and IR/Raman spectroscopies). 

### 2.2. Catalytic Evaluation: HDO with Guaiacol

Pt-Al_2_O_3_ (1 wt.%) and Ni-Al_2_O_3_ materials are catalytic supports well studied for the HDO reaction, in particular with guaiacol as a model molecule. Therefore, in this section, the discussion will be centered on the use of HPW (impregnated) or Cs_2.5_ salt (mechanical mixture) together with either Pt-Al_2_O_3_ (1 wt.%) or Ni-Al_2_O_3_ supports and a more detailed study on the effect of the formers on the catalytic hydrodeoxygenation of guaiacol. 

Figure 10 shows the evolution of conversion for all catalysts with time-on-stream (TOS). In a quick inspection, one can see that initial conversion values for Ni-Al_2_O_3_-based catalysts are much higher when compared with Pt-Al_2_O_3_ ones, probably because of the higher amount of metal present in those samples (28 against 1 wt.%). Accordingly, the evolution with TOS for both supports is very distinct, with a strong conversion decrease in the case of the Ni-Al_2_O_3_ series whereas conversion for Pt-Al_2_O_3_ catalysts series remains relatively constant over time, meaning that deactivation is more important for the Ni-Al_2_O_3_ series. This might be due to the fact that this catalysts series presents more acidity when compared with the Pt series, as demonstrated by NH_3_-TPD results. The addition of Cs_2.5_ salt does not bring any advantage, in terms of guaiacol conversion for both series, especially for Ni-Al_2_O_3_-based catalysts. 

The addition of HPW, however, leads to an increase in the initial conversion, although the deactivation is more pronounced, especially for Pt-Al_2_O_3_-based series. This might be due to the formation of carbonaceous deposits on the acid sites, as previously reported by Bohaene et al. [27]. On the other hand, it was previously reported that solids such as Al_2_O_3_ support that possess basic (OH) sites, have the tendency to decompose heteropolyacids or, at least, to interact strongly with them [15,26], which can explain the lower conversion of HPW/Pt-Al_2_O_3_ catalyst. In the case of Ni-Al_2_O_3_ support, the presence of a substantially higher metal content could attenuate this effect, explaining the higher conversion of HPW/Ni-Al_2_O_3_ sample, despite the significant decrease in conversion with TOS, which can be attributed to deactivation phenomena as this sample presents the highest acidity. Regarding the more severe deactivation verified for HPW-impregnated samples, XPS analysis showed the presence of two different heteropolyacid species dispersed onto the alumina support, probably with distinct acidity properties. Maybe in this case, HPA species presenting a higher acidity strength (such as Brönsted acid sites, see Section 2.1.4) could explain this deactivation phenomenon. 

Attending to these findings, further exploration of the catalytic results will be focused on the Ni-Al_2_O_3_ catalysts series. Concerning the product distribution for the Ni-Al_2_O_3_ support series, the main products that were identified and quantified are: phenol, cyclohexanone, anisole and benzene, summarized in Figure 11. 

Minor products, found but not quantified, are discriminated in Table 2. These, as well as other products detected in trace amounts were grouped as “others”. 

As it can be observed, the main reaction product is phenol, except for Ni-Al_2_O_3_ and HPW/Ni-Al_2_O_3_ at 10 min TOS where benzene is the main product. According to these results, it can be assumed that the main reaction pathway is the direct deoxygenation (DDO) over the hydrogenation one, once phenol is one of the major products of the reaction, which is in agreement with the literature regarding Ni immobilized onto acid supports. For instance, Zhao et al. [28] reported that DDO is the preferred pathway for this type of catalyst when low pressures are applied. This happens due to the steric effect of the methoxy group of guaiacol molecule, making guaiacol co-planar adsorption onto the support a more difficult process, leading to a direct deoxygenation pathway. The trend of the selectivity to phenol is the same for all the Ni-Al_2_O_3_-based catalysts and increases with TOS. The presence of cyclohexanone in the product pool might suggest that phenol is suffering secondary reactions. For the Ni-Al_2_O_3_ catalyst, and especially the HPW/Ni-Al_2_O_3_ one, a significant amount of benzene in the final products is detected, but this product’s selectivity decreases with time whereas phenol selectivity increases, meaning that there is probably a loss in hydrodeoxygenation capacity as some deactivation occurred, above all, for the samples with higher acidity. This is the case as these two catalysts are the ones that present the highest acidity. According to Song et al. [9], this can be attributed to a synergetic effect between Ni and acid sites in the presence of H_2_. However, the same trend was not verified for Ni-Al_2_O_3_/Cs_2.5_ catalyst. This presents a significant amount of phenol (more than 70%), especially at 120 min TOS, which can be attributed to the small amount of Cs_2.5_ salt on the catalysts, when compared with other authors [29,30], that observed the presence of deoxygenated products with catalysts loaded with a larger amount of Cs_2.5_ salt (>80%). The presence of anisole was also detected on Ni-Al_2_O_3_ and HPW/Ni-Al_2_O_3_ catalysts, the same catalysts with higher amount of benzene, meaning that those catalysts are able to eliminate the high-energy-demanding C_aromatic_-OR bonds, with R being either hydrogen or a methyl group, without ring hydrogenation. 

Concerning the “other” products, their selectivity increases with TOS, with Ni-Al_2_O_3_/Cs_2.5_ being the exception, since it remains rather constant. As the retention times for these products are longer, this might indicate that these are heavier products. In fact, the identified but not quantified products are, amongst others, 1-methoxy-3-methylbenzene and 1,2-dimethoxybenzene (see Table 2), which are heavier than phenol. 

In terms of spent catalyst characterization, it was confirmed by XRD analysis that none of the catalysts suffered the loss of structure. However, it was observed by IR spectroscopy two new bands characteristic of guaiacol. Popov et al. reported that guaiacol adsorbs on the alumina surface with both oxygens [31] and it is thought to be one of the main deactivation causes. 

## 3. Materials and Methods

### 3.1. Catalysts Preparation

The tungsten heteropolyacid, H_3_PW_12_O_40_·x H_2_O (HPW) was purchased from ThermoFischer (Kandel, Germany) GmbH, Cesium Acetate (95%) from Fluka, Platinum on Alumina (1 wt.% of Pt) from Sigma-Aldrich, PURAL SB (Al_2_O_3_, 70.3 wt%) from Condea and Nickel (II) nitrate hexahydrate (99%) from Merck.

Firstly, PURAL SB was calcined in a muffle (Nabertherm, Germany) under air at 600 °C for 5 h (heating rate of 5 °C·min^−1^). Then, a solution of 6 M of Ni(NO_3_)_2_·6H_2_O was prepared and then impregnated onto the support by using an incipient wetness impregnation method (IWI). With the volume of Ni nitrate solution used, the amount of Ni added to the support was calculated to be 28% (in weight). The resulting slurry was left to dry overnight at 120 °C. The powder obtained was afterwards treated for 30 min at 450 °C, under nitrogen (heating rate of 5 °C·min^−1^ and flow of 60 mL·min^−1^) and then under reducing atmosphere (60 mL·min^−1^ of N_2_ and 20 mL·min^−1^ of H_2_) during 4 h. 

For the HPW-impregnated catalysts, an aqueous solution of 0.01 M of HPW was prepared and then used for impregnation (IWI) onto both Pt-Al_2_O_2_ and Ni-Al_2_O_3_ supports. The materials obtained were left to dry overnight at 120 °C. The final powders were treated in an inert atmosphere with a heating rate of 5 °C·min^−1^ with a flow of 60 mL·min^−1^ of N_2_ until it reached the final temperature of 350 °C. When this temperature was reached, there was a 2 h plateau. The final HPW content was 18 wt.% for both supports.

The Cs_2.5_ salt (hereafter Cs_2.5_) was prepared according to Okuhara et al. [32], by adding the appropriate amount of the cesium precursor (CsCH_3_COO, C = 0.04 M) dropwise to an aqueous solution of 0.01 M of HPW with continuous and vigorous stirring, at room temperature. The Cs salt starts to precipitate, forming a milky-looking solution. This solution was left to age overnight and then washed several times by centrifugation (Centurion Scientific, UK-C2 Series, 4 washings 6000 rpm, with times ranging from 15 to 20 min each). The yield of this process was 69%. The powder obtained was calcined in the air in a muffle (Nabertherm, Germany) at 350 °C for 2 h (heating rate of 5 °C·min^−1^). After thermal treatment of the supports and Cs_2.5_, physical mixtures between each support and Cs salt were made. The powders were ground until a homogeneous mixture was obtained. The final Cs_2.5_ salt content was 20.5 wt.% for Pt-Al_2_O_2_ and 20 wt.% Ni-Al_2_O_3_. In the end, all catalysts were ground to 63–125 μm aggregate particle size. A summary of the catalysts can be seen in Table 3.

### 3.2. Catalyst Characterization

The PXRD patterns were recorded for all catalysts (fresh and spent) on a Bruker D8 Advanced X-ray diffractometer (Billerica, MA, USA), with Cu Kα radiation (1.5406 Å) and equipped with a 1D LYNXEYE XE detector. The measurement conditions were the following: 40 kV–40 mA, step size of 0.05° (2 Theta), step time of 1s; a Ni filter was also used to remove Cu Kβ contribution. Crystallography Open Database (COD) was used to identify the diffractograms obtained.

The IR spectra for all catalysts (fresh and spent) were obtained using a Nicolet 6700 FTIR spectrometer from ThermoScientific (Waltham, MA, USA), with wavenumbers between 4000 and 400 cm^−1^ and a 4 cm^−1^ resolution (64 scans). The technique used was transmission mode, which was then converted into absorbance. Samples were diluted (1 wt.%) in KBr and then pressed at 8 tons to get a final pellet for transmission measurements.

UV-Vis DRS spectra were obtained for Pt-Al_2_O_3_ and Ni-Al_2_O_3_ catalysts on a Varian Cary 5000 from Agilent (Santa Clara, CA, USA) within the range 200–800 nm, with a Praying Mantis (integration sphere) accessory for DRS measurements. The reflectance spectra were converted into *F(R)* through the Kubelka–Munk function: $F\left(R\right)=K/S={\left(1-R\right)}^{2}/\left(2R\right)$, with *K*, *S* and *R* being, respectively, the absorption, scattering and diffuse reflectance [33]. 

Raman spectra were collected with a Labram HR 800 Evolution equipment from Horiba, JobinYvon. Spectra were obtained with a 532 nm excitation source and the laser power at the samples was ~10 mW. Data were collected for 10 sec and 4 accumulations using a 100× objective lens.

A non-monochromatic dual anode XSAM800 spectrometer from KRATOS (Manchester, UK) was used to perform X-ray photoelectron spectroscopy (XPS) studies. The Al Kα X-rays source was used to irradiate the samples which were fixed at a take-off angle of 45° and analyzed in an ultrahigh vacuum (UHV) chamber (~10^−7^ Pa) at room temperature. X-rays were produced using a high voltage of 12 kV and a filament current of 10 mA. Spectra were acquired in FAT mode, with a pass energy of 20 eV, setting an energy step of 0.1 eV. Powdered samples were mounted on the XPS holder with double-face tape. X-ray source satellites were subtracted from spectra. Shirley backgrounds and Gaussian–Lorentzian products were used to fit experimental curves using the freeware XPSPeak4.1. No charge compensation was used. The charge accumulation was corrected, using as a reference the binding energy (BE) of aliphatic carbon atoms from contamination, set at 285 eV. The quantification factors were those of the library of Vision 2 for Windows, Version 2.2.9 from KRATOS. Different spectral features superpositions were detected. The most critical one was that of Pt 4f on Al 2p regions. For that reason, for samples where platinum and aluminum existed, Pt 4d and Al 2s were studied. In samples where both nickel and aluminum existed, Al 2s was on Ni 3s (non-discernible and, therefore, not used) and Al 2p was partially overlapping Ni 3p, but perfectly resolved. Still in these samples and in the presence of cesium, Al 2p was coincident with Cs 4d but the regions were easily distinguished upon peak fitting.

H_2_-TPR profiles for Ni catalysts were recorded on a Micromeritics AutoChem II (Norcross, GA, USA) by using a heating rate of 10 °C·min^−1^ until 900 °C. H_2_/O_2_ titration was performed on 200 mg of a reduced Pt-Al_2_O_3_ sample at 35 °C on the same equipment, with a loop volume of both H_2_ and O_2_ of 0.07 cm^3^.

N_2_ sorption measurements were carried out on a Micromeritics ASAP 2010 (Norcross, GA, USA) analyzer. Prior to N_2_ sorption, samples were outgassed under vacuum at 90 °C for 1 h and then at 300 °C for at least 4 h. 

Acidity analysis was done by NH_3_ Temperature Programmed Desorption. The samples (150 mg) were pre-treated at 350 °C for 1 h, 5 °C∙min^–1^ and then saturated with 15% NH_3_ in He for 1 h at 125 °C, following a cleaning step to remove physiosorbed NH_3_ at 150 °C for 1 h. The analysis was performed with a temperature program from 150 to 750 °C with a 10 °C∙min^–1^ heating ramp.

### 3.3. Catalytic Tests

The HDO reaction was carried out at atmospheric pressure in a vertical fixed-bed glass reactor having a porous plate to support the catalyst and in its central part a thermometric wheel in which a Ni-chromonickel thermocouple is inserted to allow the control of the catalyst temperature during the reaction. The liquid feed flow rate was 3.0 mL·h^−1^ of 5% *v/v* of guaiacol diluted in *n*-heptane by using a 776 Dosimat liquid pump system from Methrom (Herisau, Switzerland). Both liquid and H_2_ feed were introduced from the top of the reactor and the H_2_/guaiacol molar ratio was 50. The products were collected throughout time in a glass vase collector immersed in an ice bath and the reaction was stopped after 2 h (120 min). The reaction was carried out at 300 °C using 100 mg of catalyst. Prior to the reaction, the catalyst was pre-treated with H_2_ during 1 h, at the reaction conditions. The product analysis was conducted on a Perkin Elmer AutoSystem Gas Chromatograph (Perkin Elmer, Norwalk, CT, USA), with a Flame Ionization Detector (FID) using N_2_ as a carrier gas in a 30 m DB-5MS capillary column (inner diameter of 0.32 mm and an I.D. of 0.25 μm) from Agilent (Agilent Technologies, Little Falls, DE, USA) to follow the reaction’s progress. 

Ethylbenzene was used as an internal standard and added (3 wt.%) to the liquid samples after the reaction. The peak integration was made using DataApex CSW32 software 1.4 (DataApex, Prague, Czech Republic). The conversion (*x_GUA_*) was calculated as $x_{GUA}=1-(Area_{GUA}/\sum^{\;}Area_{i})\;$, the yield of the different products (*Y_P_*) and selectivity (*S_P_*) as, respectively $Y_{P}=Area_{P}/\sum^{\;}Area_{i}$ and $S_{P}=Y_{P}/x_{GUA}$. In some cases, Gas Chromatography–Mass Spectrometry (GC/MS) was carried out in a YL6900 GC/MS, with a mass spectrometric detector, from Young In Chromass (Anyang-si, Gyonggi-do, Korea) in a Rt-Q-Bond column (inner diameter of 0.32 mm and an I.D. of 0.10 μm) from Restek (PA, Bellefonte, PA, USA).

## 4. Conclusions

Heterogeneous heteropolyacid (HPA)-based catalysts (20 wt% HPA) were successfully prepared by using two distinct alumina-based supports, Pt-Al_2_O_3_ (1 wt.% Pt) and Ni-Al_2_O_3_ (28 wt.% Ni) and two different HPA precursors, phosphotungstic acid (HPW) and its derivative Cs_2.5_ salt. HPA was dispersed by using an incipient wetness impregnation method while Cs_2.5_ salt was mechanically mixed with the supports. 

Different characterization methods were used to confirm the presence of both HPA and Cs_2.5_ salt in the final solid catalysts. While the presence of Cs_2.5_ salt was easy to demonstrate, on the contrary, HPA was shown to interact strongly with both supports, making difficult the characterization of the final HPA-impregnated heterogeneous catalysts. However, XPS experiments, among other techniques, allowed us to confirm HPA presence, though with a fingerprint substantially modified. 

HDO reaction with guaiacol as an oxygenated model molecule was carried out with all the catalysts. Guaiacol conversion was, in particular, higher for the Ni-Al_2_O_3_-based catalysts when compared with Pt-Al_2_O_3_ ones, probably because of the higher amount of metal (28 wt.% Ni against 1 wt.% Pt), making these catalysts more acidic, as confirmed by NH_3_-TPD measurements. For this reason, a more detailed analysis of these nickel-based catalysts was performed. The presence of benzene, which is a fully deoxygenated product was verified for both Ni-Al_2_O_3_ and HPW/Ni-Al_2_O_3_ catalysts. Moreover, cyclohexane and cyclohexene were also identified for HPW/Ni-Al_2_O_3_. Phenol was one of the major identified compounds for all catalysts. Taking into account the nature of the products obtained, it is possible to conclude that the main reaction pathway is probably the direct deoxygenation of guaiacol. HPW-impregnated catalysts were the ones that showed the stronger deactivation with time on stream, probably because they are the catalysts, along their respective group (Pt-Al_2_O_3_ and Ni-Al_2_O_3_), that present the highest acidity. Finally, to understand better the reasons for the important catalyst deactivation, it would be interesting to study other catalysts with different (higher and lower) heteropolyacid contents. This study is in progress.

## Data Availability

Data present in this work are available in the article and Appendix A.

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
