# Peer review of "Heterogenization of Heteropolyacid with Metal-Based Alumina Supports for the Guaiacol Gas-Phase Hydrodeoxygenation"

_molecules, 2023, doi:10.3390/molecules28052245_

Round 1
Reviewer 1 Report
The manuscript describes the use of heteropolyacid supported on Pd or Ni-Alumina gas phase hydrodeoxygenation of guaiacol. The HDO of guaiacol as model of bio-oil derived from the pyrolysis of lignocellulosic biomass is an important way to produce bio-based chemicals and fuels. Revisions of the text are needed according to the points below. The authors should emphasize the suggestions in the revised manuscript.
INTRODUCTION
1 The authors should describe articles in the literature with the similar use of heteropolyacids in HDO reactions, to highlight the novelty of the present manuscript. Please, include the following articles:
-Poole, O., Alharbi, K., Belic, D., Kozhevnikova, E. F., & Kozhevnikov, I. V. (2017). Hydrodeoxygenation of 3-pentanone over bifunctional Pt-heteropoly acid catalyst in the gas phase: Enhancing effect of gold. Applied Catalysis B: Environmental, 202, 446-453.
-Kannan, S., Arumugam, P., & Govindasamy, G. (2022). SBA-15 and carbon supported nickel, heteropoly acid catalysts for the hydrodeoxygenation of lignin derived trans-anethole to sustainable aviation fuel. Journal of Porous Materials, 1-15.
RESULTS
2 In “In this case, Ni-Al2O3 support seems to possess smaller crystallites than Pt-Al2O3 one.” “This observation can be confirmed easily as Ni-Al2O3 support presents a surface area of about 215 cm2.g-1 while Pt-Al2O3 support presents a surface area of only 155 cm2.g-1.”, the authors should present more evidence about this statement or remove the phrase. The reflections of XRD pattern of alumina is not clear to state this.
3 The authors should present data from XRD, IR, Raman, DRS of HPW and Cs.25 in the main figures of the text. This will facilitate understanding the comparison between catalysts and starting materials.
4 In Fig. 7f there are two peaks of Cd in the spectra of Cs. Please, check Fig. 7.
5 In “According to them, when heteropolyacids are impregnated onto a basic oxide support (like Al2O3)”, this statement is not clear because alumina is a typical acid material, not basic oxide. The same question is valid for “On the other hand, it was previously reported that basic solids, such as Al2O3 support,”
6 Many important catalyst characterization results have been presented. The authors should compare the HDO results with characterization techniques, beyond the acidity, including conversion, and selectivity to phenol and benzene.
EXPERIMENTAL
7 In “For the HPW impregnated catalysts, an aqueous solution of 0.01 M of HPW was prepared and then used for impregnation (IWI) onto both Pt-Al2O2 and Ni-Al2O3 supports.”, please explain the method used (wet impregnation?, with filtration or drying?)
8 Table 3, please, explain how you measured the content of Ni, HPW and Cs2,5 in the catalysts.
9 In “The liquid feed flow rate was 3.0 mL·h-1”, please explain how was fed the liquid, using a pump?
CONCLUSIONS
10 If possible, the authors should present ideas to obtain a more promising HDO catalyst with less deactivation.
Author Response
REVIEWER 1:
The manuscript describes the use of heteropolyacid supported on Pd or Ni-Alumina gas phase hydrodeoxygenation of guaiacol. The HDO of guaiacol as model of bio-oil derived from the pyrolysis of lignocellulosic biomass is an important way to produce bio-based chemicals and fuels. Revisions of the text are needed according to the points below. The authors should emphasize the suggestions in the revised manuscript.
INTRODUCTION
1 The authors should describe articles in the literature with the similar use of heteropolyacids in HDO reactions, to highlight the novelty of the present manuscript. Please, include the following articles:
-Poole, O., Alharbi, K., Belic, D., Kozhevnikova, E. F., & Kozhevnikov, I. V. (2017). Hydrodeoxygenation of 3-pentanone over bifunctional Pt-heteropoly acid catalyst in the gas phase: Enhancing effect of gold. Applied Catalysis B: Environmental, 202, 446-453.
-Kannan, S., Arumugam, P., & Govindasamy, G. (2022). SBA-15 and carbon supported nickel, heteropoly acid catalysts for the hydrodeoxygenation of lignin derived trans-anethole to sustainable aviation fuel. Journal of Porous Materials, 1-15.
answer: The authors thank the reviewer for recommendations and added the suggested references in the introduction, stressing the use of HPA in HDO reactions.
RESULTS
2 In “In this case, Ni-Al2O3 support seems to possess smaller crystallites than Pt-Al2O3 one.” “This observation can be confirmed easily as Ni-Al2O3 support presents a surface area of about 215 cm2.g-1 while Pt-Al2O3 support presents a surface area of only 155 cm2.g-1.”, the authors should present more evidence about this statement or remove the phrase. The reflections of XRD pattern of alumina is not clear to state this.
answer: As we cannot, at the present, give more evidence about that specific statement, we decide to remove the sentence and replace the text by “On the other hand, raw Ni-Al2O3 support presents a similar PXRD pattern to Pt-Al2O3 one.”
3 The authors should present data from XRD, IR, Raman, DRS of HPW and Cs.25 in the main figures of the text. This will facilitate understanding the comparison between catalysts and starting materials.
answer: Authors agreed with reviewer’s comment and added to the respective figures, when necessary, the data form HPW and Cs2.5 salt compounds.
4 In Fig. 7f there are two peaks of Cd in the spectra of Cs. Please, check Fig. 7.
answer: The authors corrected the mistake in the figure 7.
5 In “According to them, when heteropolyacids are impregnated onto a basic oxide support (like Al2O3)”, this statement is not clear because alumina is a typical acid material, not basic oxide. The same question is valid for “On the other hand, it was previously reported that basic solids, such as Al2O3 support,”
answer: The authors do not agree completely with reviewer’s comment. Actually, Al2O3 oxide is a well-known amphoteric material, presenting both acidic and basic sites (as stressed in references present in the manuscript). Indeed, authors were, in the manuscript, referring to the basic character of Al2O3 support. Consequently. Authors replaced in the manuscript basic Al2O3 material by basic character or basic sites of alumina support.
6 Many important catalyst characterization results have been presented. The authors should compare the HDO results with characterization techniques, beyond the acidity, including conversion, and selectivity to phenol and benzene.
answer: The authors added some additional comparison between characterization results and samples catalytic behavior.
EXPERIMENTAL
7 In “For the HPW impregnated catalysts, an aqueous solution of 0.01 M of HPW was prepared and then used for impregnation (IWI) onto both Pt-Al2O2 and Ni-Al2O3 supports.”, please explain the method used (wet impregnation?, with filtration or drying?)
answer: In fact the information is already in the manuscript (incipient wetness impregnation, IWI), i.e. in the experimental section, second paragraph of the catalysts preparation section. By the way we reinforced the method used by changing the text from “onto the support (incipient wetness impregnation, IWI)” to “onto the support by using incipient wetness impregnation method (IWI)”.
8 Table 3, please, explain how you measured the content of Ni, HPW and Cs2,5 in the catalysts.
answer: Because both HPW and Cs2.5 salt-based catalysts have been prepared by either incipient wetness impregnation or physical mixture followed by calcination, we assumed that contents presented in Table 3 are identical to the amount of each compound (HPW or Cs2.5 salt) used to prepare the different catalysts by either IWI or mechanical mixture. We clarified this point as a note in Table 3.
9 In “The liquid feed flow rate was 3.0 mL·h-1”, please explain how was fed the liquid, using a pump?
answer: The information concerning feed system was added to the manuscript: “The liquid feed flow rate was 3.0 mL·h-1 of 5 % v/v of guaiacol diluted in n-heptane by using a 776 Dosimat liquid pump system from Methrom, Herisau, Switzerland).”
CONCLUSIONS
10 If possible, the authors should present ideas to obtain a more promising HDO catalyst with less deactivation.
answer: The authors believe a higher HPA content in the final catalyst would allow to overcome the problem of deactivation. This statement has been added in the conclusion section.
Reviewer 2 Report
The authors have presented the investigation on the preparation of Metal-Based Alumina Supported Heteropolyacid which were used for the Guaiacol Gas-Phase Hydrodeoxygenation. Heteropoly acids have long been utilized in acid catalytis, but it still hold importance in a variety of crucial organic transformation, particularly in Hydrodesulfurizations and Hydrodeoxygenation reactions. Therefore, more advanced level knowledge can facilitate the readers with existing challenges and their solutions, as well as how these heterogeneous catalysts can be modified to improve their performance for different types reactions. The manuscript is well presented with sufficient data and the materials were thoroughly characterized. Therefore, it can be published after considering few minor revisions. Such as,
1. Importance of heteropoly acids in heterogeneous catalysis should be emphasize more clearly in the introduction.
2. Few sentences about the heterogeneous catalysis, particularly with references to the supported catalysts would enhance the introduction
3. A scheme would offer better understanding of the work on the first glance.
4. Few minor language corrections are required
5. Add the following references
Alsalme, Ali, et al. "Probing the catalytic efficiency of supported heteropoly acids for esterification: Effect of weak catalyst support interactions." Journal of chemistry 2018 (2018).
Varala, Ravi, et al. "Sulfated tin oxide (STO)–Structural properties and application in catalysis: A review." Arabian Journal of Chemistry 9.4 (2016): 550-573.
Adil, Syed Farooq, et al. "Advances in graphene/inorganic nanoparticle composites for catalytic applications." The Chemical Record 22.7 (2022): e202100274.
Author Response
REVIEWER 2:
The authors have presented the investigation on the preparation of Metal-Based Alumina Supported Heteropolyacid which were used for the Guaiacol Gas-Phase Hydrodeoxygenation. Heteropoly acids have long been utilized in acid catalytis, but it still hold importance in a variety of crucial organic transformation, particularly in Hydrodesulfurizations and Hydrodeoxygenation reactions. Therefore, more advanced level knowledge can facilitate the readers with existing challenges and their solutions, as well as how these heterogeneous catalysts can be modified to improve their performance for different types reactions. The manuscript is well presented with sufficient data and the materials were thoroughly characterized. Therefore, it can be published after considering few minor revisions. Such as,
- Importance of heteropoly acids in heterogeneous catalysis should be emphasize more clearly in the introduction.
answer: The authors emphasized the use of heteropolyacids in heterogeneous catalysis, as recommended by the reviewer.
- Few sentences about the heterogeneous catalysis, particularly with references to the supported catalysts would enhance the introduction
answer: The authors also added some sentences concerning heterogeneous catalysis, as suggested by the reviewer.
- A scheme would offer better understanding of the work on the first glance.
answer: The authors added at the beginning of section 2 (results) a scheme, as suggested by the reviewer!
- Few minor language corrections are required
answer: Authors have checked for English improvement along all the manuscript.
- Add the following references
Alsalme, Ali, et al. "Probing the catalytic efficiency of supported heteropoly acids for esterification: Effect of weak catalyst support interactions." Journal of chemistry 2018 (2018).
Varala, Ravi, et al. "Sulfated tin oxide (STO)–Structural properties and application in catalysis: A review." Arabian Journal of Chemistry 9.4 (2016): 550-573.
Adil, Syed Farooq, et al. "Advances in graphene/inorganic nanoparticle composites for catalytic applications." The Chemical Record 22.7 (2022): e202100274.
answer: The authors added references form Alsalme et al. and Adil et al. However, we did not think review from Varal et al. was essential for the manuscript.